# Molecular Analyses of Clinical Isolates and Recombinant SARS-CoV-2 Carrying B.1 and B.1.617.2 Spike Mutations Suggest a Potential Role of Non-Spike Mutations in Infection Kinetics

**DOI:** 10.3390/v14092017

**Published:** 2022-09-12

**Authors:** Andrei Veleanu, Maximilian A. Kelch, Chengjin Ye, Melanie Flohr, Alexander Wilhelm, Marek Widera, Luis Martinez-Sobrido, Sandra Ciesek, Tuna Toptan

**Affiliations:** 1Institute for Medical Virology, University Hospital, Goethe University Frankfurt, D-60596 Frankfurt am Main, Germany; 2Texas Biomedical Research Institute, San Antonio, TX 78227-5302, USA; 3German Centre for Infection Research (DZIF), Partner Site Frankfurt am Main, D-60596 Frankfurt am Main, Germany; 4Fraunhofer Institute for Translational Medicine and Pharmacology ITMP, Theodor Stern Kai 7, D-60595 Frankfurt am Main, Germany

**Keywords:** SARS-CoV-2, spike protein, VoC, BAC mutagenesis, en passant mutagenesis

## Abstract

Some of the emerging severe acute respiratory syndrome coronavirus 2 (SARS-CoV-2) variants are less susceptible to neutralization with post-vaccine sera and monoclonal antibodies targeting the viral spike glycoprotein. This raises concerns of disease control, transmissibility, and severity. Numerous substitutions have been identified to increase viral fitness within the nucleocapsid and nonstructural proteins, in addition to spike mutations. Therefore, we sought to generate infectious viruses carrying only the variant-specific spike mutations in an identical backbone to evaluate the impact of spike and non-spike mutations in the virus life cycle. We used en passant mutagenesis to generate recombinant viruses carrying spike mutations of B.1 and B.1.617.2 variants using SARS-CoV-2- bacterial artificial chromosome (BAC). Neutralization assays using clinical sera yielded comparable results between recombinant viruses and corresponding clinical isolates. Non-spike mutations for both variants neither seemed to effect neutralization efficiencies with monoclonal antibodies nor the response to treatment with inhibitors. However, live-cell imaging and microscopy revealed differences, such as persisting syncytia and pronounced cytopathic effect formation, as well as their progression between BAC-derived viruses and clinical isolates in human lung epithelial cell lines and primary bronchial epithelial cells. Complementary RNA analyses further suggested a potential role of non-spike mutations in infection kinetics.

## 1. Introduction

Generating recombinant viruses using reverse genetics approaches serves as a powerful tool to study the biology of viral infections [1,2,3]. It helps to understand the mechanisms of virus life cycle, transmission and pathogenesis, as well as to identify viral and host factors involved in these processes. In addition, recombinant viruses can be designed to express reporter genes to facilitate cell-based screenings for antivirals and to monitor the efficacy of therapeutic and preventive countermeasures [4,5,6,7,8]. 

In response to the ongoing coronavirus disease 2019 (COVID-19) pandemic, several groups have developed infectious complementary DNA (cDNA) clones for its etiological agent, severe acute respiratory syndrome coronavirus 2 (SARS-CoV-2), using reverse genetics systems, including molecular cloning, subgenomic amplicons [9], in vitro ligation followed by electroporation [10], and circular polymerase extension reaction (CPER) [11]. Manipulation of such reverse genetics systems to introduce deletions, insertions of reporter genes and targeted mutations can be laborious and time consuming. Another commonly used model is based on the cloning of large genomes as bacterial artificial chromosomes (BACs) [12,13] followed by genome manipulation using red recombination, which is a homologous recombination-based technique used for genetic engineering in bacteria [13,14]. BACs have been developed for human coronaviruses (CoVs), such as OC-43, SARS-CoV, MERS-CoV [2,15,16], and recently for SARS-CoV-2 [17,18]. 

In this study, we describe the application of en passant mutagenesis [19] to generate recombinant SARS-CoV-2 with spike mutations from different variants of concern (VoC) strains. These emerging variants carry sets of mutations in spike but also non-spike region including accessory and non-structural genes that can potentially alter virus characteristics and lead to increased transmission rate, disease severity, risk of reinfection, susceptibility to treatment, and escape immunity [20,21,22,23,24,25,26]. The B.1 variant with D614G spike mutation became dominant in June 2020 and spread globally [27]. Similarly, the B.1.617.2 (Delta) variant emerged in April 2021 [28] and quickly displaced other variants in multiple countries. The delta variant is almost twice as transmissible as the alpha [29] and more likely to break the protection afforded by vaccines and prior infections with other variants. Numerous reports have indicated that the B.1.617.2 variant exhibits reduced sensitivity to monoclonal antibodies [30,31,32], and thus may lead to severe disease. Therefore, in this pilot study we focused our analysis on these two variants. Neutralizing capabilities of therapeutic monoclonal antibodies, convalescent or vaccine-induced sera have been evaluated by different groups using either pseudoviruses or clinical isolates [30,33,34,35]. However, by generating a fully infectious virus carrying exclusively the relevant spike mutations compared to the parental isolate, we sought to evaluate the impact of these spike mutations alone in the virus life cycle and protective effects of neutralizing and monoclonal antibodies in vitro.

## 2. Materials and Methods

### 2.1. Cell Culture and Clinical SARS-CoV-2 Isolates

Human colon adenocarcinoma (Caco-2) [36] and human lung adenocarcinoma A549 overexpressing ACE2 and TMPRSS2 (A549-AT) [37] cells were cultured in Minimum Essential Medium (MEM) supplemented with 10% fetal calf serum (FCS), 2 mM L-glutamine, 100 IU/mL of penicillin and 100 µg/mL of streptomycin at 37 °C, 5% CO_2_. All culture reagents were purchased from Sigma (St. Louis, MO, USA). The Caco-2 cells were originally obtained from DSMZ (Braunschweig, Germany, no.: ACC 169), differentiated by serial passaging and selected for high permissiveness to virus infection [36,38,39].

SARS-CoV-2 clinical isolates (CI) B.1^CI^ (MT358643) [38] and B.1.617.2^CI^ (MZ315141) [30] were isolated from nasopharyngeal patient swabs and propagated by infecting Caco-2 cells in MEM supplemented with 1% FCS as previously described [38] and early passage virus stocks were stored at −80 °C. For virus titration, A549-AT or Caco-2 cells were seeded one day prior to infection at a density of 3.5 − 4 × 10^4^ cells/well. The virus was titrated by ten-fold serial dilutions to obtain 50% tissue culture infective dose (TCID_50_) and using end-point assay until 4 days post infection (dpi) as previously described [38]. In addition, we performed spike staining to validate the virus titers (Appendix A). Cells were fixed with acetone-methanol mix (40:60) and blocked in 2% bovine serum albumin (BSA), 5% FCS and 0.01% thimerosal in washing buffer (50 mM Tris, 150 mM NaCl, pH 7.5). Following incubation with anti-SARS-CoV-2 spike antibody (40150-R007, SinoBiological, Beijing, China, dilution 1:1000) for 1 h at 37 °C, the wells were rinsed with washing buffer, incubated with secondary antibody (AffiniPure Goat Anti-Rabbit IgG, 111-035-144, Jackson ImmunoResearch, Ely, UK, dilution 1:1000) for 1 h at 37 °C and followed by treatment with AEC (3-amino-9-ethylcarbazole) solution containing 0.03% H_2_O_2_. Spike positive area was scanned and colorimetric quantification of the object counts was performed using Biosys Bioreader^®^-7000 F-z (Biosys, Karben, Germany). The mean value from uninfected wells was used for background signal normalization. All infections were carried out in a BSL-3 laboratory. Sample inactivation for further processing was performed with previously evaluated methods [39]. 

Human bronchial epithelial cells HBEpC (Promocell) were expanded using PneumaCult^TM^-Ex Plus Medium (Stemcell Technologies, Vancouver, CO, Canada) supplemented with PneumaCult^TM^-Ex Plus 50 × Supplement (Stemcell Technologies, Vancouver, CO, Canada) and after one further passage were differentiated into a pseudostratified air-liquid interface (ALI) culture in Costar^®^ 12 mm Transwell^®^, 0.4 µm Pore Polyester Membrane Inserts (Stemcell Technologies, Vancouver, CO, Canada), according to manufacturer’s instructions. For differentiation, PneumaCult^TM^-ALI Basal Medium (Stemcell Technologies, Vancouver, CO, Canada) supplemented with PneumaCult^TM^-ALI 10 × Supplement (Stemcell Technologies, Vancouver, CO, Canada), PneumaCult^TM^-ALI 100 × Maintenance Supplement (Stemcell Technologies, Vancouver, CO, Canada), 0.2% heparin solution (Stemcell Technologies, Vancouver, CO, Canada), 200 × hydrocortisone stock solution (Stemcell Technologies, Vancouver, CO, Canada), and 0.1 mg/mL Primocin^®^ (Invivogen, San Diego, CA, USA) was used, according to manufacturer’s instructions. For culture maintenance, medium was exchanged at least once a week and apical compartment was washed once a week for 10 min with 1 × PBS (Life Technologies, Carlsbad, CA, USA). Cells were incubated at 37 °C, 5% CO_2_ until fully differentiated, i.e., cilia movement and mucus secretion were visually detectable and permanent.

### 2.2. Generation of Recombinant SARS-CoV-2 by En Passant Mutagenesis

For the generation of recombinant viruses, we used the SARS-CoV-2 BAC construct based on the USA-WA1/2020 strain [17], which is named as WT^BAC^ in this study. We generated two different SARS-CoV-2 BAC mutants that carry the variant-specific spike regions of clinical isolates B.1^CI^ (FFM7 isolate) [38] and B.1.617.2^CI^ [30] using en passant mutagenesis [19,40]. In brief, cDNA was generated using SARS-CoV-2 variant RNA using SuperScript VILO Kit (Thermo Fisher Scientific GmbH, Bremen, Germany). Individual spike regions (~4 kb) were amplified using Phusion Polymerase Kit (New England Biolabs, Frankfurt, Germany) with specific primers (P1 and P2) (Appendix A). Amplified spike regions were cloned into a pMiniT shuttle vector (New England Biolabs, Frankfurt, Germany) and analyzed by Sanger sequencing. A PCR product comprising of a kanamycin cassette, an I-*Sce*I restriction site and a 50-bp duplication of the spike gene was amplified using P3 and P4 primer pair (Appendix A) and pEPkan-S2 plasmid DNA as a template (Addgene #61601) [19].The amplicon was inserted into the *Nco*I single cutter site within the pMiniT-spike gene (pMiniT-spike-kana). For en passant mutagenesis WT^BAC^ was transformed into *E. coli* GS1783 cells, which contain a temperature-dependent expression cassette for the recombination proteins and an L-arabinose-inducible I-*Sce*I enzyme [40]. Subsequently 100 ng of NotI-digested spike-kanamycin cassette of the pMiniT-spike-kana construct, was electroporated into competent GS1783 cells with WT^BAC^ to allow the first step of recombination. Following restriction digestion analysis, the recombinant clones were subjected to second en passant step to remove the kanamycin selection marker following L-arabinose treatment and I-*Sce*I expression. Genome integrity and the excision of the kanamycin cassette was verified by restriction digestion analysis (Appendix A) and Sanger sequencing. 

### 2.3. Transfection and Reconstitution of Recombinant Viruses

BAC DNA was isolated using NucleoBond Xtra Midi Kit (Macherey Nagel, Düren, Germany). Caco-2 cells were transfected with 2.5 μg of BAC DNA using TransIT-LT1 transfection reagent (Mirus, Madison, WI, USA), according to manufacturer’s instructions in MEM supplemented with 10% FCS (6-well plates, triplicates). Next day, medium was exchanged with 1% FCS-MEM and cells were incubated at 37 °C, 5% CO_2_ for another 3–5 days until the cytopathic effects (CPEs) emerged. When the CPEs reached to 80%, flasks were transferred to −80 °C to facilitate cell lysis and virus release. Supernatant was harvested, clarified by centrifugation at 1000× *g* for 10 min and stored at −80 °C. Virus titers were determined by end-point titration (TCID50/mL) using Caco-2 and A549-AT cells as described above (Appendix A).

### 2.4. Nanopore Sequencing and Bioinformatic Analysis

cDNA and library preparation reactions were prepared following the ARTIC SARS-CoV-2 sequencing protocol (https://www.protocols.io/view/ncov-2019-sequencing-protocol-v3-locost-bh42j8ye, last accessed on 6 June 2022) using V4 primer scheme (Integrated DNA Technologies, Coralville, IA, USA), LunaScript RT SuperMix Kit and Q5 Hot Start High-Fidelity 2× Master Mix (New England Biolabs, Frankfurt, Germany) (https://github.com/artic-network/primer-schemes/tree/master/nCoV-2019/V4, accessed on 6 June 2022). Pooled and cleaned PCR reactions were quantified using Qubit dsDNA HS Assay Kit (Thermo Fisher Scientific GmbH, Bremen, Germany). The Ligation Sequencing kit, Native Barcoding Expansion kit (EXP-NBD104 (SQK-LSK109; Oxford Nanopore Technologies, England), NEB Blunt/TA Ligase Master Mix, NEBNext Ultra II End repair/dA-tailing Module and NEBNext Quick Ligation (New England Biolabs, Frankfurt, Germany) were used to prepare the library following the manufacturer’s protocol. The barcoded library was quantified by Qubit dsDNA HS Assay Kit (Thermo Fisher Scientific GmbH, Bremen, Germany) and R9.4 flowcell (FLO-MIN106; Oxford Nanopore Technologies) were primed and loaded with 20 ng of library. Sequencing was performed on an MK1c instrument (Oxford Nanopore Technologies) for 8 h with high-accuracy basecalling (Guppy, v5.0.17) [41]. Consensus sequences were built from the barcode-sorted, quality-filtered FAST5 and FASTQ files, using the ARTIC pipeline (https://artic.network/ncov-2019/ncov2019-bioinformatics-sop.html, accessed on 6 June 2022). Medaka algorithm was used for variant calling. Reads were aligned to the NC_045512.2 reference using minimap2 [42] and visualized using IGV (v2.8.13) [43] and Geneious Prime Software (Biomatters Ltd., Auckland, New Zealand). Mutations identified in recombinant and clinical variants are listed in Appendix A.

### 2.5. Infection Kinetics by Cell Confluence Measurement and Live Cell imaging

A549-AT cells were seeded in 96-well plates (3.5 × 10^4^/well) one day prior to infection with SARS-CoV-2 clinical isolates B.1^CI^, B.1.617.2^CI^, and recombinant viruses B.1^BAC-V^, B.1.617.2^BAC-V^ and WT^BAC-V^ at a multiplicity of infection (MOI) of 1. The virus dilutions were calculated using a conversion formula: PFU (mL)/TCID_50_ (mL) = 0.7 [44]. Cells were incubated in a Spark Cyto 400 multimode plate reader with cell imager and environmental control module (Tecan Group Ltd., Zürich, Switzerland) at 37 °C, 5% CO_2_, for 46 h. Cell confluence measurements and imaging were performed at 1 h intervals with 4× magnification. Non-infected controls (0 h time point) were used for the normalization representing 100% confluency. Automated monitoring was performed in three biological replicates and confirmed by bright-field microscopy. Back titration of each virus solution was performed in parallel via end-point titration using A549-AT cells to confirm equal amounts of infectious particles (Appendix A). Statistical analysis was performed by comparing every time point with multiple two-way *t*-tests (Appendix A), standard deviation within the replicates is indicated by the error bars. 

### 2.6. Neutralization Assays Using Vaccine Sera

We used a total of 28 sera collected two-weeks after either three doses of mRNA vaccine BNT162b2 (BNT, Comirnaty™, BioNTech/Pfizer, Mainz, Germany/NYC, U.S.) (3 × BNT, n = 16) or a combination scheme with two doses of mRNA-1273 (MOD, Spikevax™, Moderna, MA, USA) and one booster dose of BNT162b2 (2 × MOD + BNT booster, n = 12) (Appendix A). SARS-CoV-2 antibody concentrations were determined using the SARS-CoV-2 IgG (Nucleocapsid) and II Quant assay (spike RNA receptor binding domain (RBD)) kits with an analytical measurement as Index > 1.4 and 2.98 – 5680 binding antibody units per mL (BAU/mL), respectively, in an Alinity I device (Abbott Diagnostics, Wiesbaden, Germany) (Appendix A). For the neutralization assays, 3.5 × 10^4^ cells were seeded in 96-well plates one day prior to infection. Then, heat inactivated sera (56 °C for 30 min) were serially diluted (1:2) in MEM supplemented with 1% FCS, and incubated with 400 TCID_50_ of the indicated SARS-CoV-2 clinical isolate or recombinant virus in a total volume of 100 μL (37 °C for 1 h). Medium with virus was aspirated and replaced with 100 μL fresh MEM with 1% FCS. As controls, we used a serum obtained before the onset COVID-19 pandemic and infected cells without any serum. Back titration of each virus solution was performed in parallel by diluting the working solution serially to a final concentration of 1:1000 (1 × 10^−4^) and using A549-AT cells to confirm equal amounts of infectious particles (Appendix A). CPEs were evaluated microscopically 48 h post infection (hpi). The 50% neutralization titer (NT_50_) for each serum tested was determined by normalizing to control infections. Experiment was carried out twice, yielding similar results and each serum was tested in two biological replicates per run.

### 2.7. Neutralization Assay Using Therapeutic Monoclonal Antibodies

A549-AT cells (3.5 × 10^4^/well, 96-well plate) were seeded in 96-well plates one day prior to infection. Monoclonal antibodies (Casirivimab (REGN10933), Imdevimab (REGN10987), and a combination of both (REGN-COV)) were diluted serially in a 1:3 ratio starting with a working concentration of 3000 ng/mL in MEM supplemented with 1% FCS and subsequently infected with clinical isolate and recombinant viruses at an MOI of 0.1 at 37 °C, 5% CO_2_. At 48 hpi, cells were fixed using 3% paraformaldehyde (PFA) in PBS. 50% neutralization titer (NT_50_) was determined by brightfield microscopy in terms of cell confluency using Spark Cyto 400 multimode plate reader (Tecan Group Ltd., Zürich, Switzerland). Infected and non-infected wells containing no antibodies were used as positive and negative control, respectively, to determine relative confluency. Each monoclonal antibody was tested in three biological replicates. The 50% neutralization concentration of the employed mAbs was determined by normalizing to infection controls. 

### 2.8. Antiviral Testing Using Small-Molecule Inhibitors

A549-AT cells (3.5 × 10^4^/well, 96-well plate) were seeded in 96-well plates one day prior to infection. On the following day, cells were treated with Remdesivir, Nafamostat mesylate, Nirmatrelvir (PF-07321332), and Molnupiravir (EIDD-2801) (MedChem Express, Monmouth Junction, NJ, USA) with a starting concentration of 3000 ng/mL and 1:3 dilutions to a final concentration of 0.3 ng/mL and subsequently infected with indicated viruses at an MOI of 0.1 for 48 h. The cells were fixed in a 3% PFA-PBS and analyzed by confluency measurement using Spark Cyto 400 multimode plate reader as described in the previous section. Infected and non-infected wells containing no antiviral drug were used as positive and negative control to determine relative confluency. The 50% maximal inhibitory concentration (IC_50_) was normalized to control infections and used to determine the inhibitory potency of the employed drug. 

### 2.9. Infection of HBEpC and TEER Measurement 

Prior to infection, complete medium was applied to apical compartment and incubated for 2 h. For the transepithelial electrical resistance (TEER measurement), Millicell^®^ ERS-2 Volt-Ohm electrode (Merck) was disinfected and equilibrated in 0.1 M KCl before measurement. Next, cells were washed three times with 1x PBS for 10 min each. HBEpC were infected with recombinant viruses and corresponding clinical isolates at an MOI of 1 by applying virus-containing medium to the apical compartment. In total, 2 hpi, apical medium was aspirated and cells were washed three times with 1x PBS for 10 min each. Final washing step was collected as an input supernatant for qRT-PCR. Basal compartment media were exchange by fresh growth medium and cells were incubated at 37 °C, 5% CO_2_. In total, 5 dpi, medium was applied to apical compartment and incubated for 2 h. Following TEER measurement, 5 dpi supernatants were collected and stored at −80 °C. 

### 2.10. Infection Kinetics by RNA Analysis

A549-AT cells (4.5 × 10^4^/well, 96-well plate) were infected with clinical isolate (B.1^CI^, B.1.617.2^CI^) and recombinant (B.1^BAC-V^, B.1.617.2^BAC-V^, WT^BAC-V)^ viruses at MOI of 1 in 1% MEM. After 1 hpi, virus containing medium was aspirated, cells were rinsed three times with PBS and supplemented with fresh 1% MEM. Diluted virus inoculum (0 hpi), cell lysates and supernatants were harvested 2, 4, 6, 8, 10, and 12 hpi using RNeasy 96 QIAcube HT Kit (Qiagen, Hilden, Germany) and QIAamp 96 Virus QIAcube HT Kit (Qiagen, Hilden, Germany), respectively, according to manufacturer’s protocol. Back titration of each virus dilution was performed in parallel via end-point titration using A549-AT cells to confirm equal amounts of infectious particles (Appendix A). 

RNA from infected A549-AT and HBEpCs was extracted and purified using the QIAcube HT (Qiagen, Hilden, Germany), according to the manufacturer’s protocol. For the RNA standard curve, in vitro transcribed N gene RNA was generated using HiScribe^TM^ T7 High Yield RNA synthesis kit (New England Biolabs, Frankfurt, Germany), according to manufacturer’s instructions. Amplicons were generated from SARS-CoV-2 cDNA using N-specific primers (Appendix A, P14 and P15) following the Q5^®^ High-Fidelity PCR Kit (New England Biolabs, Frankfurt, Germany), according to manufacturer’s protocol. Concentration of synthesized RNA was determined by UV/VIS spectrometry at λ = 260 nm. Based on transcript size, copy numbers/mL were calculated and transcript was diluted accordingly. Purified and in vitro transcribed RNA samples were subjected to qRT-PCR using Reliance One-Step Multiplex qRT-PCR Supermix (Bio-Rad, Hercules, CA, USA), primers and probes specific to N gene (P8–P10) and subgenomic (sg) N-gene RNA (P11–P13) (Appendix A). Ct values of cellular genes were normalized to RNase P expression (P5-P7) in infected and uninfected samples. qRT-PCR was performed in a C1000 touch thermal cycler (Bio-Rad, Hercules, CA, USA). Thermal cycling was performed at 50 °C for 10 min, followed by 95 °C for 10 min (initial denaturation) and 40 cycles of 95 °C for 10 s (denaturation) and 60 °C for 30 s (annealing and elongation). Concentrations of sub-genomic and total N-gene were determined using N-gene standard curve calculations. Standard deviation between three replicates was indicated through error bars.

### 2.11. Western Blot Analysis

For preparation of protein extracts, HBEpC were lysed using RIPA buffer (150 mM NaCl, 50 mM Tris/HCl pH 8.0, 1% (*v*/*v*) Triton X-100, 0.5% (*v*/*v*) sodium deoxycholate, 0.3% (*v*/*v*) SDS, 5 mM NaF, 1 mM Na_3_VO_4_, 1x cOmplete Mini, EDTA-free protease inhibitor cocktail (Merck), 2 mM MgCl_2_ and 1 U/mL Pierce^TM^ Universal Nuclease (Thermo Scientific). Protein concentrations were determined using DC Protein Assay Kit (Bio-Rad, Hercules, CA, USA) samples were separated by SDS-PAGE and were blotted onto Amersham^TM^ Protran^TM^ Premium 0.45 µm nitrocellulose membrane (Cytiva, Marlborough, MA, USA) for 1.5 h using const. 120 V. Membranes were blocked for 1 h in 5% (*w*/*v*) BSA in TBS-t (150 mM NaCl, 20 mM Tris, 0.1% (*v*/*v*) Tween-20) and incubated with SARS-CoV-2 (2019-nCoV) Spike S1 Antibody (1:1000, #40150-R007, Sino Biological, Beijing, China), SARS-CoV/SARS-CoV-2 Nucleocapsid antibody (1:1000, #40143-MM05, Sino Biological, Beijing, China) and β-actin antibody (1:1000, #4967, Cell Signaling Technology, Leiden, The Netherlands) overnight at 4 °C in blocking buffer. Secondary antibodies IRDye^®^ 800CW goat anti-rabbit IgG (H + L) and IRDye^®^ 680RD goat anti-mouse IgG (H + L) (LI-COR Biosciences GmbH, Bad Homburg, Germany) were diluted 1:40.000 in blocking buffer and incubated for 1 h at room temperature. Blots were imaged using CLx imaging device (LI-COR Biosciences GmbH, Bad Homburg, Germany).

### 2.12. Statistical Analysis

Statistical analyses were performed using GraphPad Prism 9 (GraphPad Software, San Diego, CA USA). Results were considered statistically significant when *p* < 0.05.

For infection kinetics, mean values of confluence measures of BAC and respective CI measured using Spark Cyto 400 multimode plate reader (Tecan) were compared individually for each time point by multiple two-way *t*-tests (Appendix A). TCID_50_ values were compared using one-way analysis of variance (ANOVA) We furthermore compared vRNA levels in A549-AT cells at different time points using Welch *t*-test (Appendix A). NT_50_ results of the recombinant variant and respective clinical isolate obtained by microscopical analysis were compared through one-way (ANOVA). Dose-responses for monoclonal antibodies and antiviral compounds non-linear regression method was used. Regression was performed on 6 and 3 independent data points for mAb treatment and inhibitor treatment, respectively. Derived IC_50_ values are depicted in the respective graph. Using the same software, unpaired *t*-tests were performed for statistical analysis. Multiple comparisons were corrected by using the incorporated Bonferroni–Dunn method. For analyses with HBEpCs one-way ANOVA was used to determine statistically significant differences.

### 2.13. Ethics Statement 

The study was conducted according to the guidelines of the Declaration of Helsinki, and approved by the Institutional Review Board of the Ethics Committee of the Faculty of Medicine at Goethe University Frankfurt (2021-201, 20-864 and 250719). 

## 3. Results

### 3.1. Generation of Recombinant SARS-CoV-2 Carrying B.1 and B.1.617.2 Spike Mutations

The main aim of this work was to generate recombinant SARS-CoV-2 variants using the same virus backbone by en passant mutagenesis in order to study the impact of variant-specific spike mutations during infection. The entire spike coding regions of SARS-CoV-2 B.1 and B.1.617.2 clinical isolates (CI, ~3.9 kb) were amplified and cloned into a shuttle vector (Figure 1A). Within the spike region, a PCR amplicon comprising a kanamycin expression cassette, an I-SceI homing endonuclease site and a 50 bp spike duplication was inserted (Figure 1A). The transfer of NotI-linearized spike-kanamycin cassette (~5 kb) was mediated by electroporation of GS1783 cells with WT^BAC^ during the first recombination step. Recombinant clones were screened by restriction digestion and subsequently were subjected to the second en passant step. The arabinose-mediated expression of I-SceI enzyme induced a double strand break in the BAC, which facilitated the homologous recombination between the 50 bp genome duplications and seamless excision of the kanamycin resistance marker (Figure 1B). Restriction digestion analysis confirmed the genome integrity and the excision of the selection marker (Appendix A). Recombinant viruses WT^BAC-V^, B.1^BAC-V^ and B.1.617.2^BAC-V^ were reconstituted in Caco-2 cells and the supernatants were harvested and used for an additional infection round to generate passage one stocks. Nanopore sequencing of recombinant viruses and clinical isolates confirmed genome integrity and mutations (Figure 1C, Appendix A).

### 3.2. Recombinant SARS-CoV-2 Variants B.1^BAC-V^ and B.1.617.2^BAC-V^ Show Different Growth Kinetics Than Corresponding Clinical Isolates

To assess the infection kinetics of recombinant BAC-derived viruses in comparison to corresponding clinical isolates, we performed live-cell imaging of infected A549-AT cells and monitored the syncytia, CPE formation and cell confluency over the course of 46 h (Figure 2, Appendix A). Label-free, automated cell confluence assessment of BAC-derived viruses and clinical isolates infected cells were mostly comparable up until 8 hpi (Figure 2A, Appendix A). Automated confluency measurement of the infected cells showed significant differences between BAC-derived viruses (B.1^BAC-V^ in green and B.1.617.2^BAC-V^ in orange) and the clinical isolates (B.1^CI^ and B.1.617.2^CI^ in grey) (Figure 2A). Cell-lysis related confluency reduction seems to be delayed by 4 h in cells infected with clinical isolates (Figure 2A). Based on TCID_50_ calculation_,_ viral particles in supernatant at 24 hpi do not significantly differ between recombinant viruses and clinical isolates. Nevertheless, we observed slightly higher means for titers of clinical isolates. (Figure 2A, right). In all conditions, syncytia were formed between 8–10 hpi (Appendix A). We observed nearly full-lysis of B.1^BAC-V^ and WT^BAC-V^ infected cells 14–15 hpi (Figure 2B, Appendix A). B.1^CI^ and B.1.617.2^BAC-V^ infection mediated cell lysis was observed 17–18 hpi (Figure 2B, Appendix A). Syncytia in B.1.617.2^CI^ persisted longer than other conditions and full cell lysis was detected at approximately 21–22 hpi (Figure 2B, Appendix A ).

We also analyzed the replication kinetics by measuring total and sub-genomic RNA expression for N-gene using qRT-PCR. A549-AT cells were infected with indicated recombinant and clinical variants at an MOI of 1. Supernatant (SN) and cells were harvested at given time points. In vitro transcribed N-gene RNA template was used as a standard for absolute quantification of the viral RNA in infected cells (Figure 3, Appendix A). The time point of 0 hpi indicates RNA levels in the inoculum used for the infection. Over the course of 12 h, infection kinetic total and sub-genomic N-gene RNA levels in B.1^BAC-V^ and WT^BAC-V^ infected cells were comparable indicating a similar entry efficiency and replication kinetic (Figure 3A,B top, green and blue, respectively). In contrast, RNA copy levels in B.1^CI^ inoculum and at 2 hpi were 1-2 log less abundant (Figure 3A,B top, grey; Appendix A). Yet it gradually matched to B.1^BAC-V^ and WT^BAC-V^ levels by 8 hpi suggesting a more efficient replication kinetic. Total and subgenomic N-gene levels in B.1.617.2^CI^ and B.1.617.2^BAC-V^ infected cells were lower, but showed a steeper increase over the course of infection and matched to WT^BAC-V^ by 12 hpi (Figure 3A,B bottom, orange and grey). Similar to intracellular RNA levels, total N gene levels in the supernatant were comparable in WT^BAC-V^ and B.1^BAC-V^ (Figure 3C, top, green and blue) and levels in B.1^CI^ started to increase at 6 hpi and matched the levels of WT^BAC-V^ and B.1^BAC-V^ by 8 hpi (Figure 3C, top, grey). RNA levels in B.1.617.2^BAC-V^ supernatant were comparable to WT^BAC-V^ by 8 hpi (Figure 3C, bottom, orange and blue) and B.1.617.2^CI^ showed a steep increase at later time points (Figure 3C, bottom, grey).

### 3.3. Vaccine and Convalescent Sera Conduct Similar Neutralization of BAC-Derived Viruses and Corresponding Clinical Isolates

To assess the sensitivity of recombinant and clinical variants to serum antibodies (Appendix A), we used samples from triple-vaccinated individuals with mRNA vaccines (3× BNT or 2× MOD + BNT booster). Serum dilutions and respective virus were pre-incubated for 1 h before applying the mixture to confluent A549-AT cells, incubated for 48 h and neutralization titers subsequently determined microscopically. The 50% neutralization titer (NT_50_) was determined, using the half-maximal inhibitory concentration values of serum samples, normalized to control infections, from their serial dilutions. In our analysis, NT_50_ of 3x BNT vaccine sera were comparable between recombinant viruses and clinical variants (Figure 4A). In contrast to B.1.617.2^BAC-V^, we observed a significant increase in NT_50_ for B.1^BAC-V^ compared to the corresponding clinical isolate, while using sera from individuals who received 2x MOD + BNT booster (Figure 4B).

### 3.4. Treatment of Recombinant Viruses and Clinical Isolate with Monoclonal Antibodies and Small Molecule Inhibitors

Anti-Spike monoclonal antibodies (mAbs) provide useful prophylactic interventions for SARS-CoV-2 infection. In this assay, we tested mAbs Imdevimab, Casirivimab, individually and combined. To that end, A549-AT cells were incubated with mAbs at different concentrations and indicated viruses for 48 h at an MOI of 0.1 (Figure 5). Dose-response was monitored by automated brightfield microscopy. Overall, neutralization upon Casirivimab and combination (Casirivimab + Imdevimab) treatment was comparable for recombinant viruses and the corresponding clinical isolates. Despite minor significant differences (Casirivimab-B.1: 10, 30 ng/mL; B.1.617.2: 30 ng/mL, Casirivimab + Imdevimab-B.1.617.2: 10, 30 ng/mL), calculated NT_50_ values for these treatments remain similar. The treatment with Imdevimab displayed diminished protection against both recombinant variants, B.1^BAC-V^ and B.1.617.2^BAC-V^, showing significantly lower inhibitory potential between 10 and 300 ng/mL (B.1: 10, 30, 100, 300 ng/mL; B.1.617.2: 30, 100, 300 ng/mL). This is underlined by 2.4- and 3.2-fold lower NT_50_ values, respectively, when comparing to the corresponding clinical isolates.

We further performed comparative analysis to assess response to antivirals (Figure 6). A549-AT cells were incubated with Remdesivir, Nafamostat mesylate, Nirmatrelvir, and Molnupiravir and indicated viruses using an MOI of 0.1. Dose-response was monitored by brightfield microscopy in terms of CPE development at 48 hpi. Antiviral efficacy of all tested inhibitors was comparable against recombinant viruses and clinical isolates as indicated by IC_50_ values.

To assess the replication dynamics of recombinant SARS-CoV-2 and clinical isolates in human air-liquid interphase (ALI) cultures, fully differentiated human bronchial epithelial cells (HBEpC) were infected with an MOI of 1 (Figure 7). Five dpi infected HBEpCs were analyzed for morphological features, barrier integrity, and RNA expression. We observed more pronounced CPE formation with B.1.617.2^BAC-V^ and B.1.617.2^CI^ compared to other viruses tested. WT^BAC-V^ showed the weakest CPE among all (Figure 7A). In order to test the barrier function, we measured the transepithelial electrical resistance (TEER) between electrodes placed in the apical and basal compartments of reconstructed bronchial epithelia. In line with the marked CPE development, impairment of the epithelial barrier was more prominent for B.1.617.2^CI^, showing a 3.4-fold reduction in TEER (Figure 7B). Remaining viruses showed 1.5-fold to 2.2-fold reductions, similar to uninfected control (Mock). RNA analyses were performed using cellular extracts and showed a significant difference between B.1.617.2^CI^ and B.1.617.2^BAC-V^ but not for B.1 clinical isolate and recombinant variant. We also detected a significant increase of extracellular viral RNA (** *p* < 0.005) and infectious viral particles (*** *p* < 0.001) based on the TCID_50_ values for B.1.617.2^CI^ (Figure 7C). Concordantly we observed a slightly elevated expression of viral Spike and Nucleocapsid proteins in cells infected with B.1^CI^ and B.1.617.2^CI^ compared to those infected with recombinant counterparts (Figure 7D).

## 4. Discussion

In this study we sought to investigate the role of spike and non-spike mutations in virus life cycle by comparing recombinant SARS-CoV-2 viruses and their parental clinical isolates. During live-cell imaging we observed that infected A549-AT cells, independent of the virus used, developed syncytia at approximately 8 hpi (Appendix A, Figure 2A). Syncytia are formed by the fusion of surface-exposed spike on infected cells to ACE2 of adjacent cells [45]. Expression of spike protein alone without other viral proteins in vitro can trigger this formation, to a lesser degree, has been observed for other CoVs, such as SARS-CoV and MERS-CoV [46,47]. Multinucleated giant cell formation and cellular syncytia were also observed in 87% of post-mortem lung tissues expressing SARS-CoV-2 RNA and S proteins, underscoring the importance of this morphological feature in COVID-19 pathogenesis [48]. However, whether syncytia formation has any relative significance to the SARS-CoV-2 infection remains unknown. Syncytia can facilitate cell-to-cell spread of the virus and shield it from neutralizing antibodies leading to immune evasion [49]. In addition, syncytia-mediated apoptosis or pyroptosis can release virus to infect neighboring cells and/or trigger an inflammatory response. In this work, we observed that the morphology of the syncytia and resulting CPEs were comparable among recombinant viruses and the corresponding clinical isolates. Interestingly, we noted that syncytia persisted longer leading to delayed CPE onset in cells infected with clinical isolates when compared to BAC-derived virus infected A549-AT cells. In primary HBEpCs, WT^BAC-V^, B.1^CI^, B.1^BAC-V^ showed comparable results in every parameter tested (Figure 7). However, we observed more pronounced CPE development for B.1.617.2^CI^ (Figure 7A), which also showed a significantly impaired barrier function when compared to B.1.617.2^BAC-V^ (Figure 7B), indicating the potential role of non-spike mutations on tissue integrity. One important caveat here is that we used differentiated HBEpCs from a single donor. Our observations might be donor-specific and this needs to be addressed in follow up studies.

In order to find out whether this delay has any relevant consequences for the virus life cycle, we performed RNA analysis. Although the RNA levels in the inoculum seem to slightly differ between CI and BAC-V (Appendix A), back titration of the viruses showed that similar infectious particles were used (Appendix A). This difference is possibly due to presence of unpacked viral RNA intermediates released into the supernatant upon lysis. On the basis of total N-gene (intracellular and supernatant) qRT-PCR, we detected significantly faster replication kinetic in cells infected with BAC-Vs at early time points (Appendix A). Despite lower inoculum, RNA levels in CI-infected cells matched the BAC-V at 6–8 hpi time point. This correlates with the RNA levels detected in the supernatant (Figure 3). Since at 12 hpi time point the syncytia are still intact, it can be assumed that total N-gene RNA measured in the supernatant is mostly due to exocytosis and can therefore reflect virus production. Although not significant, we observed a tendency of increased progeny virus titers in A549-AT cells infected with the clinical isolates, especially B.1.617.2^CI^, at 24 hpi (Figure 2A, right). Primary HBEpCs infected with B.1.617.2^CI^ showed significantly higher RNA levels in cell lysates and supernatant. Consistent with these observations, we detected higher infectious particles in the supernatant of the cells infected with, B.1.617.2^CI^ at 5 dpi (Figure 7C). On the basis of these findings, we can suggest stronger CPE in HBEpCs and longer persisting syncytia in A549-AT infected with B.1.617.2^CI^ may allow prolonged virus replication. Based on concordant observations from two different cell models, we carefully suggest a potential role of non-spike mutations in infection kinetics by either modulating syncytium formation and/or virus replication. In future studies, gene-by-gene molecular mapping of non-spike mutations by generating recombinant viruses would be necessary to prove these observations.

The D614G mutation has been claimed to increase viral fitness by enhancing virus replication and the virion stability in human lung epithelial and Calu-3 cells but not in Vero cells [50]. By comparing WT^BAC-V^ and B.1^BAC-V^ we were able to assess this in A549-AT cells and found no significant difference in terms of viral RNA expression, syncytia, and CPE development (Figure 2 and Figure 3, Appendix A). 

In functional assays, we detected minor differences between clinical isolates and recombinant viruses. In general, the treatment of recombinant viruses or clinical isolates carrying the same spike mutations displayed similar neutralization efficacy using vaccine sera. Triple-vaccinated individuals had strong neutralization responses to both tested variants (Figure 4). 3xBNT response for both variants are comparable for clinical isolates and corresponding BAC variants. In contrast, B.1^BAC-V^, when compared to B.1^CI^, was neutralized more efficiently by 2x MOD + BNT (Figure 4B). Since the spike protein is the major target of vaccine-elicited antibodies tested in this study, the difference we observed in the NT assay is unlikely to be due to the non-spike mutation P323L in B.1^CI^. The superior breadth and potency of neutralizing antibodies due to heterologous vaccine combination might play a role. However, the sample size used for this experiment was limited, and therefore it is not possible to draw any concrete conclusion. 

In line with our previous study [30] we observed a more efficient protection against the clinical isolates with the combination mAb treatment compared to single treatments with Casirivimab and Imdevimab. However, our data also indicates significantly lower protection around NT_50_ against B.1^BAC-V^ and B.1.617.2^BAC-V^ when A549-AT cells were treated with Imdevimab (Figure 5). In an earlier study, we showed that among the three mAbs, Imdevimab confers the lowest protective effect against diverse SARS-CoV-2 variants [30]. This is probably due to differential infection kinetics reflected by delayed CPE development for the clinical isolates. Recombinant viruses seem to induce CPEs and release virus earlier than clinical isolates which may lead to exhaustion of antibodies due to earlier surge of virus titers in the supernatant. Casirivimab and combination treatment are more potent and seem to counteract this effect. 

For studies with small molecule inhibitors, we tested Remdesivir and Molnupiravir (EIDD-2801) as prodrugs of the nucleosides GS-441524 and EIDD-1931, respectively (Figure 6). Both active compounds incorporate into the complementary RNA leading to lethal mutagenesis by the insertion of copious mutations into the newly synthesized RNA genome [51]. In comparison to the BAC-derived viruses, the clinical isolate B.1^CI^ carries the P323L and B.1.617.2^CI^ on top of that carries the G671S mutations within the RdRp (NSP12) (Figure 1A, Appendix A). Although molecular dynamic simulation for P323L mutation revealed a higher binding affinity to Remdesivir [52], our results did not show any difference in inhibition. In addition, we tested Nirmatrelvir, the active component of Paxlovid and a potent inhibitor of the viral 3-chymotrypsin-like protease (3CLpro), which is essential for polyprotein cleavage and generation of SARS-CoV-2 structural and non-structural proteins. Upon treatment, we observed no significant difference in dose-responses to CI and BAC-V of B.1 and B.1.617.2 variants. These results are supported by the fact that BAC-Vs and CIs harbor identical 3CLpro coding sequences. Ultimately, we considered Nafamostat mesylate as the only host-factor inhibitor targeting the TMPRSS2 protease ensuring proteolytic activation of the spike protein and facilitating entry into human airway cells [53], but we did not observe any significant difference. 

Direct comparison of clinical isolates and recombinant viruses carrying variant-specific spike gene mutations allowed us to precisely investigate the functional relevance of spike and non-spike mutations in the context of SARS-CoV-2 infection. As a proof-of-principle, we only compared spike and non-spike mutations of B.1 and B.1.617.2. However, this method can be easily implemented to generate recombinant viruses to study relevant and emerging mutations. Our results indicate that non-spike mutations do not have significant impact on the neutralization efficacy when using post-vaccine sera, mAbs, and small molecule inhibitors. Nevertheless, CPE, barrier integrity, and RNA analyses for B.1.617.2^CI^ and B.1.617.2^BAC-V^ in A549-AT cells and HBEpC suggest a potential role for the non-spike mutations of the clinical isolate in replication and infection kinetics. 

## Figures and Tables

**Figure 1 viruses-14-02017-f001:**
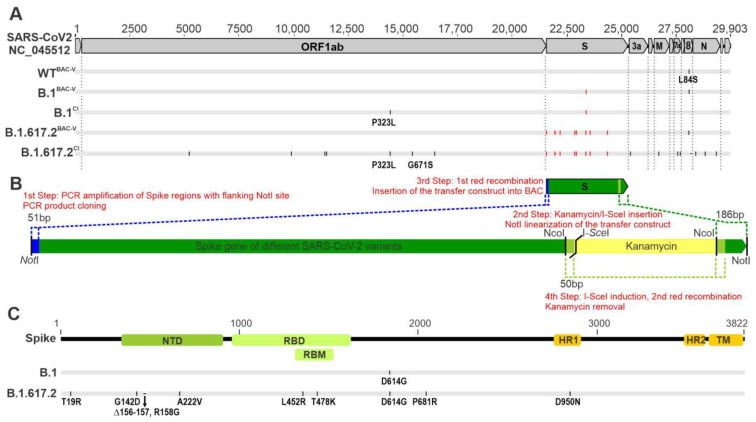
Generation of recombinant SARS-CoV-2 mutants using en passant mutagenesis. (**A**). Genome organization of SARS-CoV-2 Wuhan isolate (NC_045512) is depicted on the top. Open reading frames (ORF) are shown in grey boxes. Dotted horizontal lines mark the ORF borders. cDNA BAC clone of the SARS-CoV-2 USA-WA1/2020 isolate carrying a single amino acid substitution in ORF8 (L84S) and is represented as WT^BAC-V^. P314L and G662S in ORF1b corresponds to P323L and G671S, respectively, in RNA-dependent RNA polymerase (RdRp). Below, the clinical isolates (B.1^CI^ and B1.617.2^CI^) and corresponding BAC derived variants (B.1^BAC^ and B1.617.2^BAC^) generated in this study are shown. Genome wide mutations are indicated in red (spike region) and black (outside the spike region) short lines and are positioned according to the Nanopore sequencing results. (**B**). 1st Step: Spike regions from clinical isolate B.1 and B.1.617.2 (green box) were amplified and cloned into a shuttle vector with flanking NotI restriction sites. 2nd Step: The kanamycin cassette, I-SceI restriction site flanked by 50 bp genomic duplications (light green) were amplified and cloned into the NcoI restriction site within the spike region. NotI-digested transfer construct (~5 kb) was subsequently used for electroporation. 3rd Step: During the first red-recombination step a minimum of 51 bp (N-terminus, blue dotted lines) and 186 bp (C-terminus, green dotted lines) homolog sites facilitated the insertion of the transfer construct into the WT^BAC^ genome. 4th Step: In the second step of Red recombination I-SceI homing endonuclease expression by arabinose induction, mediated double-strand break and the seamless removal of the kanamycin cassette, resulting in BAC mutants carrying the corresponding spike mutations for B.1 and B.1.617.2 isolates. (**C**). SARS-CoV-2 spike protein subunits and domains are shown. N-terminal domain (NTD), receptor binding domain (RBD), receptor binding motif (RBM), heptad repeat 1 and 2 (HR1, HR2) and transmembrane domain (TM). Variant-specific mutations are indicated below.

**Figure 2 viruses-14-02017-f002:**
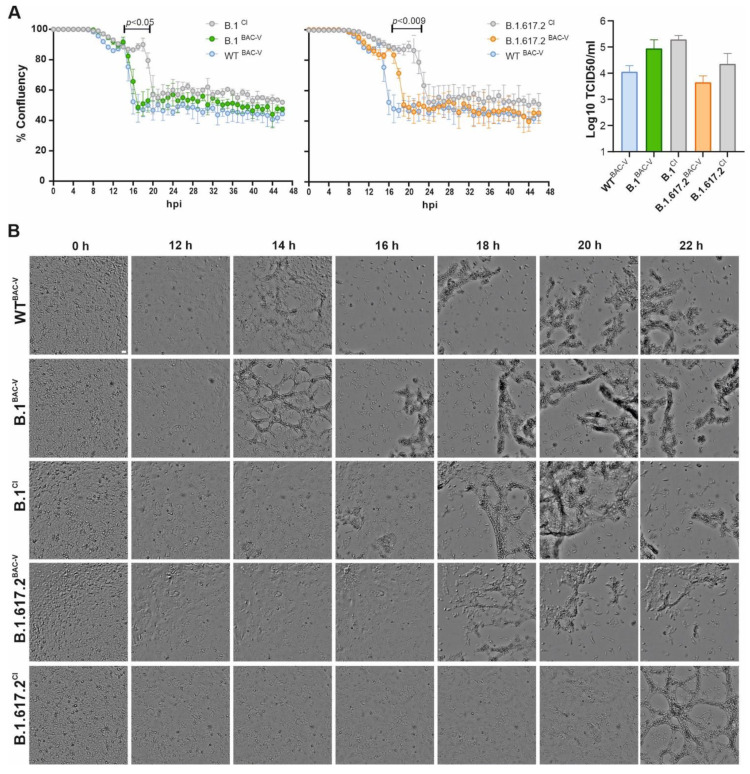
Growth kinetics of clinical and recombinant SARS-CoV-2 variants. (**A**). Graphical presentation of % confluence of infected A549-AT cells (MOI = 1) over time. For normalization, confluence of uninfected control was set to 100%. Measurements were performed every 1 h. Standard deviations derive from three technical replicates and represent two independent biological replicates. Graphs show means ± s.d.; *p* values were obtained by two-tailed unpaired *t*-tests (Appendix A), time points showing a significant difference between infection with BAC derived virus (BAC-V), and clinical isolates (CI) are indicated. Virus titer in supernatant at 24 hpi are depicted for WT^BAC-V^ (blue), B.1^BAC-V^ (green), B.1^CI^ (grey) B.1.617.2^BAC-V^ (orange), and B.1.617.2^CI^ (grey). Data are represented as mean with SD of three and four biological replicates. Conditions were compared using One-Way ANOVA revealing no significant differences. The experiment was performed twice yielding similar results. Hours post infection (hpi). (**B**). Bright-field microscopic images at 4× magnification represent the cytopathic changes observed during the course of infection at indicated time points. Scale bar: 100 µm. See also Appendix A.

**Figure 3 viruses-14-02017-f003:**
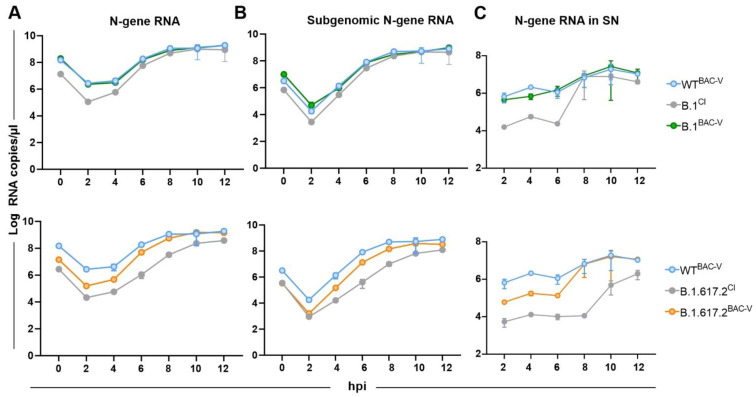
Time course analysis of viral RNA levels in A549-AT cells. (**A**). Total and (**B**). Subgenomic N-gene RNA copies in cells infected with WT^BAC-V^ (blue), B.1^CI^ (grey, top), B.1^BAC-V^ (green), B.1.617.2^CI^ (grey, bottom), and B.1.617.2^BAC-V^ (orange) at an MOI of 1. The 0 hpi stands for RNA levels in the inoculum used for the infection. (**C**). Total N-gene RNA copies in the supernatant of the cells infected with WT^BAC-V^ (blue), B.1^CI^ (grey, top), B.1^BAC-V^ (green), B.1.617.2^CI^ (grey, bottom), and B.1.617.2^BAC-V^ (orange) at an MOI of 1. Data are mean ± s.d. of three biological replicates. Statistical comparisons were performed using Welch *t*-test (Appendix A). The experiment was repeated twice yielding similar results. Hours post infection (hpi). Supernatant (SN).

**Figure 4 viruses-14-02017-f004:**
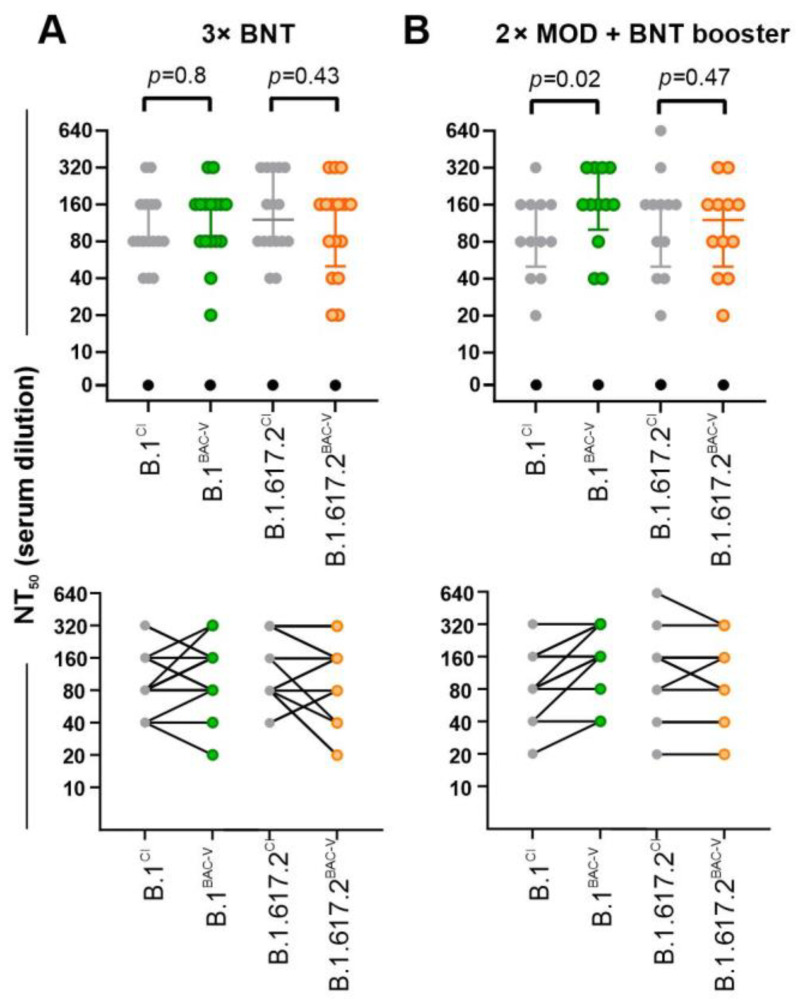
Antibody-mediated neutralization efficacy against recombinant and clinical SARS-CoV-2 variants. Graphs represent serum dilutions resulting in 50% neutralization (NT50) of B.1^CI^ (grey), B.1^BAC-V^ (green), B.1.617.2^CI^ (grey), and B.1.617.2^BAC-V^ (orange). Assays were performed using sera from individuals either (**A**). vaccinated and boosted with BNT162b2 (n = 16) (3x BNT) or (**B**). vaccinated with mRNA-1273 (MOD) and boosted with BNT162b2 (n = 12) (2x MOD + BNT booster). Negative control serum is indicated in black. Graphs represent data as median with interquartile range. One-Way ANOVA compared CI and BAC-V (*p*-values are indicated above each data set) Bottom graphs indicate relations between NT_50_ positive sera from (**A**,**B**). tested on BAC derived viruses (BAC-V) or corresponding clinical isolates (CI).

**Figure 5 viruses-14-02017-f005:**
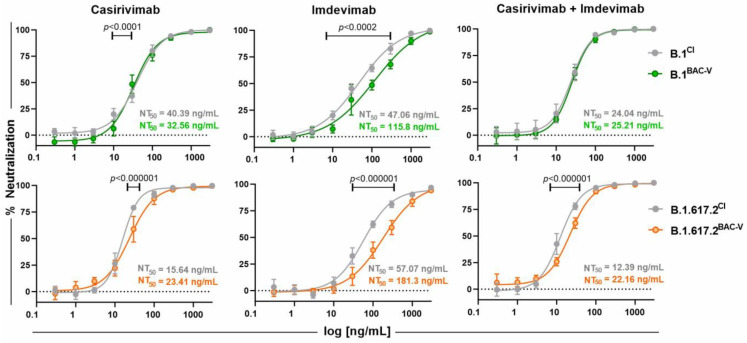
Efficacy of therapeutic mAbs against SARS-CoV-2 B.1^BAC-V^ (green), B.1.617.2^BAC-V^ (orange) and their corresponding clinical isolate (CI, grey). A549-AT cells were infected with indicated viruses at an MOI of 0.1 in the presence of mAbs (Casirivimab, Imdevimab or combination of both) at given concentrations. The % neutralization was determined by confluence measurement 48 hpi using Spark Cyto 400 multimode plate reader. Displayed data were normalized to controls containing only virus but no antibodies. The 50% neutralization (NT_50_) values were calculated by performing non-linear regression on the mean of six replicates indicated by the error bars. Single data points were compared using multiple unpaired *t*-tests. Significant results are indicated by respective *p* values. The experiment was repeated three times yielding similar results.

**Figure 6 viruses-14-02017-f006:**
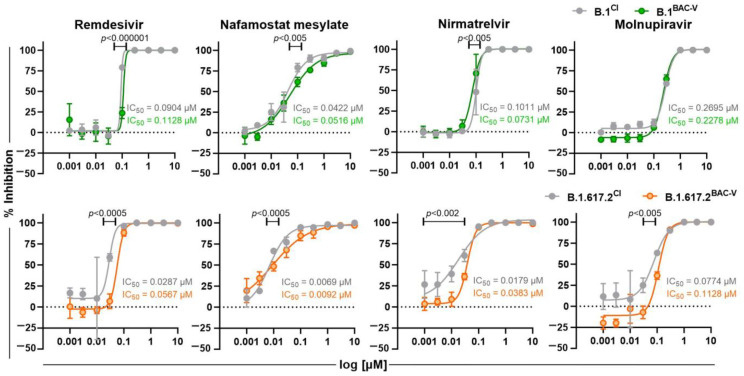
Efficacy of small molecule inhibitors Remdesivir, Nafamostat mesylate, Nirmatrelvir, and Molnupiravir against B.1^BAC-V^ (green), B.1.617.2^BAC-V^ (orange), and corresponding clinical isolates (CI, grey). Cells were infected at an MOI of 0.1 with indicated viruses. Graphs represent the % inhibition of infection determined by confluency measurement 48 hpi using Spark Cyto 400 multimode plate reader with cell imager and environmental control module. Displayed data were normalized to controls containing only virus but no inhibitors. IC_50_ values were calculated by performing non-linear regression on the mean of three biological replicates indicated by the error bars. Single data points were compared using multiple unpaired *t*-tests. Significant comparisons are indicated by respective *p* values. The experiment was repeated twice yielding similar results.

**Figure 7 viruses-14-02017-f007:**
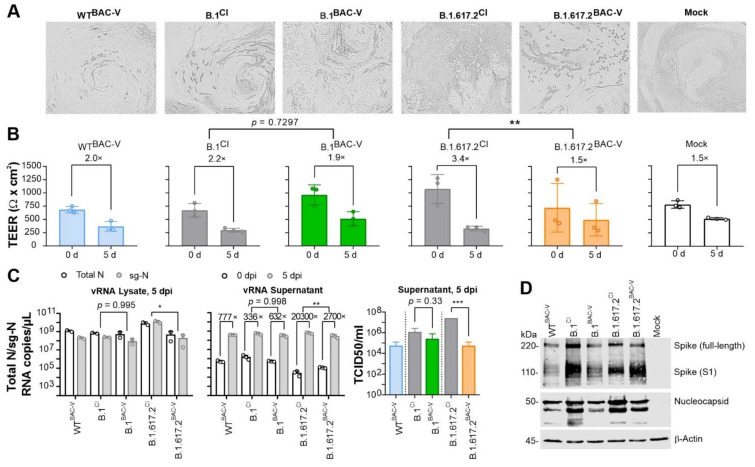
Infection of HBEpCs with recombinant and clinical SARS-CoV-2. (**A**). Brightfield microscopy of infected HBEpCs 5 dpi. (**B**). Epithelial barrier integrities upon infection with WT^BAC-V^ (blue), B.1^CI^ (grey), B.1^BAC-V^ (green), B.1.617.2^CI^ (grey), B.1.617.2^BAC-V^ (orange) and in uninfected control (colorless) 0 hpi and 5 hpi. Comparison of BAC-V and CI was performed using One-Way ANOVA. Error bars represent mean and SD of three biological replicates. ** *p* < 0.01. (**C**). RNA copies/µL of intracellular vRNA, in terms of total N (white) and sg-N (grey) after 5 dpi (left), and extracellular total N vRNA (middle). Changes of RNA levels from 0 dpi (white) to 5 dpi (grey) are indicated by x-fold for supernatants. RNA levels were calculated using N standard RNA. Comparison of BAC-V and CI RNA levels for intracellular and for changes in extracellular RNA levels was performed using One-Way ANOVA. Error bars represent mean and SD of intra- and extracellular RNA comprising three and two biological replicates, respectively. * *p* < 0.05, ** *p* < 0.005. Virus titers in supernatant 5 dpi are indicated for WT^BAC-V^ (blue), B.1^CI^ (grey), B.1^BAC-V^ (green), B.1.617.2^CI^ (grey) and B.1.617.2^BAC-V^ (orange) (right). Error bars represent mean and SD of three biological replicates. Statistical comparison of BAC-V and CI was performed using unpaired *t*-test. *** *p* < 0.001. (**D**). Western blot analysis for spike and nucleocapsid expression in HBEpC infected with indicated viruses. β-actin detection served as a loading control.

## Data Availability

Not applicable.

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
