# Peer review of "Molecular Analyses of Clinical Isolates and Recombinant SARS-CoV-2 Carrying B.1 and B.1.617.2 Spike Mutations Suggest a Potential Role of Non-Spike Mutations in Infection Kinetics"

_viruses, 2022, doi:10.3390/v14092017_

Round 1

Reviewer 1 Report

Major comments

Authors here analyzed the impact of non spike mutations of SARS-CoV-2 that emerged in Europe (B1) and India (B1.617.2) using their BAC reverse genetic system. Seroneutralizations and antivirals assays were also performed to decipher the impact of such mutations. The study is very interesting but some clairifications and confirmation must be done do improve the quality of this study.    

Firstly, in this study, the authors generated recombinant viruses carrying the spike mutations of different variant strains. In this way, they turn off all non-spike mutations but without paying attention to the structural, non-structural proteins or accessory proteins. Instead of turn off all of these mutations together, why didn't they focused on a particular gene and turn off mutations gene by gene, or at least structural and non-structural gene? It is already established to date that some specific accessory genes or non structural proteins are involved in the viral pathogenesis using reverse genetic or not (PMID : 35896528, 35562044, 35677080)...

Why didn't they generated recombinant virus carrying only non-spike mutations to compare abut more importantly to confirm all the results described in this study ?  Is this BAC reverse system user-friendly enough to achieve this?

Secondly, no animal experiments were performed to validate their in vitro results particulary for the antivirals molecules for which, during early the pandemic, controversial effects where observed in vitro vs in vivo...

Finally and more importantly, as you will see in figure 2, when authors generated a recombinant virus carrying only spike mutation (from B1 or B1.617.2), the fitness of the recombinant virus is better than the clinical strain! How can authors explain this? To date, there is no evidence that mutation decrease the viral fitness. This should be clarified.

Minor comments

Line 39 : Please, add a reference.

Line 46 : Authors stated that in vitro ligation is a time consumming process. However, the utilisation of large genomic system such as BAC can be toxic for bacteria and therefore also laborious. How many transfection attempts were performed to generate recombinant virus ?

Line 59 : reference 13 : Mélade et al., didn’t used BAC system in their study but instead cDNA fragments from de novo synthetic genes. Reference should be moved upside in line 44.

Line 61 : Please, add a reference.

Line 86 : Table S1 : Please, clarify what represent the colorimetric values inside the table (relative value I suppose).

Line 126 :  table S2 not S1

Line 144 : Until how many days before the first CPE were observed ?

Line 294 : figure 1B describing the reverse genetics methodoloy must be improve. Honestly, without the text, it's not user-friendly Maybe authors can indicate next to each step what occur (1st step : amplification of spike proteins; 2nd step : insertion into vector shuttle etc...). Also, at the end, please indicate how the kanamycin cassette is removed and what happen to the "mutated" amplified spike fragment (in which backbone the mutated spike is insert, I suppose inside WT BAC-V). Indicate the recombinant which was produce (their names for example). Need improvement.

Line 311 : same comment for line 144 : until how many days ?

Line 313 : table S3 : confusing. Please indicate which mutations appeared unintentionally : during cell culture or after transfection for example (such as L84S for the WT-BAC-V).  

Line 336 : I feel really confused. The author replace the spike of the Wuhan strain by the one of B1. When they did it, they observed a higher viral fitness of the recombinant virus compare to the clinical B1 isolate itself? How is this possible ? The mutation in Table S3 that emerged P314L (I suppose because it’s not indicated) could affect this fitness ? Does the non-spike mutation negatively impact the viral fitness ? This results must be confirmed with the generation of a recombinant virus carrying only the non spike mutation !

Line 362 : figure 3 : please indicate where are the significant differences in the figure.

Line 401 : figure 4 : my previous comment also come for here. For the 3x BNT results, NT50 for clinical and recombinant B1 or B1.617.2 were comparable. So we could speculate that the non spike mutation may not be relevant. However, this should be confirm by the respective recombinant virus carrying the non-spike mutation and confront to this result.  For the 2x MOD+BNT booster, the NT50 was higher for B1 BAC : what is the conclusion here? Does the non spike protein are involved and interfere with the neutralization ? Authors should at least discuss about this in the discussion section. From here, the authors didn't used the WT-BAC-V anymore. Why ? How can we be certain that the mutation recorded on this backbone (L84S) also present in the recombinant viruses was not involved in these testing?

Line 464 : figure 7 : Not easy to read. All this graph can be place in a same pannel. This could help the reader to see/compare the different between mock-infected and infected cells.

Line 575 : in lines 423-424 authors claimed that "treatment with Imdevimab displayed diminished protection against both recombinant variants, B.1BAC-V and B.1.617.2BAC-V". This mean that non spike mutation play a role. Since the two mAB used here targeted the spike RDB, authors should clairified the implication of the non spike mutation in the neutralization

Author Response

Response to reviewer 1

Comments and Suggestions for Authors

Major comments

Authors here analyzed the impact of non spike mutations of SARS-CoV-2 that emerged in Europe (B1) and India (B1.617.2) using their BAC reverse genetic system. Seroneutralizations and antivirals assays were also performed to decipher the impact of such mutations. The study is very interesting but some clairifications and confirmation must be done do improve the quality of this study.    

Firstly, in this study, the authors generated recombinant viruses carrying the spike mutations of different variant strains. In this way, they turn off all non-spike mutations but without paying attention to the structural, non-structural proteins or accessory proteins. Instead of turn off all of these mutations together, why didn't they focused on a particular gene and turn off mutations gene by gene, or at least structural and non-structural gene?

Response 1:

We concur that it is critical to investigate mutational landscape of other structural and non-structural proteins and its role in virus life cycle. In this current study we rather focused on the spike region due to its importance for therapeutic interventions, vaccine design and efficacy. Based on our findings we now have started looking into the role of Nucleocapsid and RdRp regions using en passant mutagenesis. As the reviewer suggests, we seek to understand the role of individual mutations or the mutations in concert for virus replication gene by gene, especially for more recent VoCs. However, due to the pace of research and publication rate in this field, we preferred sharing our results with the community in a timely manner.

It is already established to date that some specific accessory genes or non structural proteins are involved in the viral pathogenesis using reverse genetic or not (PMID : 35896528, 35562044, 35677080)...

Why didn't they generated recombinant virus carrying only non-spike mutations to compare abut more importantly to confirm all the results described in this study ?  Is this BAC reverse system user-friendly enough to achieve this?

Response 2:

It is very likely that spike mutations in concert with non-spike mutations influence the viral fitness. En passant mutagenesis can be used to introduce single/multiple mutations and large gene swabs. As briefly mentioned above, we are undertaking these steps to study relevant mutations in different gene regions, and would like to focus on replication dynamics rather than neutralization and antiviral treatments for this purpose. We therefore think that these additional analyses with recombinant viruses carrying non-spike mutations would be out of scope of this current work. However, we included the following sentences in the discussion (page 16, lines 619-621).

“…In future studies, gene-by-gene molecular mapping of non-spike mutations by generating recombinant viruses would be necessary to prove these observations…”

We appreciate though your bringing these studies to our attention. We now included the PMID 35562044 and 35677080 to the references and edited the text accordingly (Page 2, Lines 54-57)

“…These emerging variants carry sets of mutations in spike but also non-spike region including accessory and non-structural genes that can potentially alter virus characteristics and lead to increased transmission rate, disease severity, risk of reinfection, susceptibility to treatment and escape immunity [20-26]…”

Secondly, no animal experiments were performed to validate their in vitro results particulary for the antivirals molecules for which, during early the pandemic, controversial effects where observed in vitro vs in vivo...

Response 3:

We agree that he execution of animal experiments is powerful for the confirmation of in vitro results especially for drug and vaccine efficacy studies for which we did not observe any difference between recombinant viruses and the clinical isolates. Due to time und budget constraints, we decided to confirm infection kinetic results obtained from cell line models (Caco2 and A549) in primary human bronchial epithelial cells.

On further note, we carefully chose antivirals that have proven effects in vivo and in vitro. Paxlovid and Molnupiravir (EUA) are antivirals specifically developed for the treatment of COVID-19 during the pandemic. Remdesivir (FDA-approved) is used as a positive control in most compound identification studies and furthermore was in clinical use in the early phases of the pandemic. The effectiveness of Nafamostat mesylate against SARS-CoV-2 in vivo and in vitro was published multiple times as well. Keeping in mind that we only sought to rule out potential differences in neutralization efficacy caused by particular non-spike mutations, in vitro experiments are more cost-/time-efficient and deliver reproducible results.

Finally and more importantly, as you will see in figure 2, when authors generated a recombinant virus carrying only spike mutation (from B1 or B1.617.2), the fitness of the recombinant virus is better than the clinical strain! How can authors explain this? To date, there is no evidence that mutation decrease the viral fitness. This should be clarified.

Response 4:

This is a justified question and should be clarified. Viral fitness is a very broad term that is defined by a variety of parameters. In cell culture experiments like the ones we performed, those parameters would primarily comprise the replication speed and the production of mature progeny as well as the formation of cytopathic effects (CPE). In fact, the early development of strong CPE, what we are referring to in Figure 2, do not always coincide with equal amounts of virus being produced after a certain time post infection. It is rather important how long this CPE is persisting and when the cells lyse to release progeny. That is why we did not attempt to draw conclusions about differences in viral fitness. Our explanation is based on syncytium formation upon infection which enables viral cell-to-cell spread. Since all viruses tested show comparable CPE morphology but recombinant viruses induce cell lysis earlier, we would argue that this serves the production of less viral progeny by less efficient cell-to-cell spread, which is also reflected by lower virus titers detected for BAC viruses in A549-AT and primary human bronchial epithelial cells.

Minor comments

Line 39 : Please, add a reference.

Response 1: We added the following references supporting our statement. (Page 1, line 38)

  1. Kirui, J.; Freed, E.O. Generation and validation of a highly sensitive bioluminescent HIV-1 reporter vector that simplifies measurement of virus release. Retrovirology 2020, 17, 12, doi:10.1186/s12977-020-00521-5.
  2. Nogales, A.; Avila-Perez, G.; Rangel-Moreno, J.; Chiem, K.; DeDiego, M.L.; Martinez-Sobrido, L. A Novel Fluorescent and Bioluminescent Bireporter Influenza A Virus To Evaluate Viral Infections. J Virol 2019, 93, doi:10.1128/JVI.00032-19.
  3. Shang, B.; Deng, C.; Ye, H.; Xu, W.; Yuan, Z.; Shi, P.Y.; Zhang, B. Development and characterization of a stable eGFP enterovirus 71 for antiviral screening. Antiviral Res 2013, 97, 198-205, doi:10.1016/j.antiviral.2012.12.010.
  4. Zou, G.; Xu, H.Y.; Qing, M.; Wang, Q.Y.; Shi, P.Y. Development and characterization of a stable luciferase dengue virus for high-throughput screening. Antiviral Res 2011, 91, 11-19, doi:10.1016/j.antiviral.2011.05.001.
  5. Chou, S.; Van Wechel, L.C.; Lichy, H.M.; Marousek, G.I. Phenotyping of cytomegalovirus drug resistance mutations by using recombinant viruses incorporating a reporter gene. Antimicrob Agents Chemother 2005, 49, 2710-2715, doi:10.1128/AAC.49.7.2710-2715.2005.

Line 46 : Authors stated that in vitro ligation is a time consumming process. However, the utilisation of large genomic system such as BAC can be toxic for bacteria and therefore also laborious. How many transfection attempts were performed to generate recombinant virus ?

Response 2: SARS-CoV-2 BAC constructs propagated in E.coli (DH10B and GS1783) did not show any sign of  toxicity or hamper the bacteria growth. We obtained in general >50% recombination efficiency during each en passant step either for insertion of large fragments or PCR products with point mutations into the BAC genome. During the virus reconstitution step, each transfection attempt yielded in CPEs and progeny recombinant viruses which were confirmed by Nanopore sequencing.

Line 59 : reference 13 : Mélade et al., didn’t used BAC system in their study but instead cDNA fragments from de novo synthetic genes. Reference should be moved upside in line 44.

Response 3: We thank the reviewer for pointing this out. We moved the citation to Page 2, line 43-45

“…reverse genetics systems including molecular cloning, subgenomic amplicons [9], in vitro ligation followed by electroporation [10], and circular polymerase extension reaction (CPER) [11]…”

Line 61 : Please, add a reference.

Response 4: We added the following reference. (Page 2, line 60)

  1. The Scientific Pandemic Influenza Group on Modelling, Operational sub-group (SPI-M-O). Consensus statement on COVID-19, 2nd June 2021. https:// assets. publi shing. servi ce. gov. uk/ gover nment/ uploa ds/ system/ uploa ds/ attac hment_ data/ file/ 993321/ S1267_ SPI-M-O_Consensus_Statement.pdf

“…The delta variant is almost twice as transmissible as the alpha [29] and more likely to break the protection afforded by vaccines and prior infections with other variants...”

Line 86 : Table S1 : Please, clarify what represent the colorimetric values inside the table (relative value I suppose).

Response 5: We agree that the former description was not sufficient to interpret the table. In order to clarify we edited the main text (Page 3, lines 94-97) and the Table S1 in the supplemental file (Page 2 lines 8-15)

Main text:

“…Spike positive area was scanned and colorimetric quantification of the object counts was performed using Biosys Bioreader®-7000 F-z (Biosys, Karben, Germany). The mean value from uninfected wells was used for background signal normalization…”

Supplemental file:

“…Table S1. TCID50 end-point assay using spike staining in infected Caco2 and A549-AT cells. Spike positive area was scanned and colorimetric quantification of the object counts was performed using Biosys Bioreader®-7000 F-z. Uninfected wells were used for background signal normalization. Negative values following normalization were designated as zero. Viruses were diluted as indicated above the plate layout and the cells were infected in quadruplicates (A-D). Wells with infected cells displaying CPEs/syncytia (green) were positive for Spike protein expression, unless the monolayer is lysed. Higher dilutions that lack CPEs (red) did not show any Spike signal…”

Line 126 :  table S2 not S1

Response 6: Thank you for pointing out this typo. We now corrected it in the text. (Page 2, line 129)

Line 144 : Until how many days before the first CPE were observed ?

Response 7: We edited the text and included this information. (Page 4, lines 142-147)

“…Caco-2 cells were transfected with 2.5 μg of BAC DNA using TransIT-LT1 transfection reagent (Mirus, Madison, WI, USA) according to manufacturer’s instructions in MEM supplemented with 10% FCS (6-well plates, triplicates). Next day medium was exchanged with 1% FCS-MEM and cells were incubated at 37°C, 5% CO2 for another 3-5 days until the cytopathic effects (CPEs) emerged. When the CPEs reached to 80%, flasks were transferred to -80°C to facilitate cell lysis and virus release…”

Line 294 : figure 1B describing the reverse genetics methodoloy must be improve. Honestly, without the text, it's not user-friendly Maybe authors can indicate next to each step what occur (1st step : amplification of spike proteins; 2nd step : insertion into vector shuttle etc...). Also, at the end, please indicate how the kanamycin cassette is removed and what happen to the "mutated" amplified spike fragment (in which backbone the mutated spike is insert, I suppose inside WT BAC-V). Indicate the recombinant which was produce (their names for example). Need improvement.

Response 8: We agree that it would be easier for the readers to be able to follow the recombination steps from the figure itself without the need of the legend-text. Therefore, we included step 1-4 in Figure 1 (Page 8, line 337) and included brief explanations for the steps used to introduce the VoC spike cassettes.

We also added in the figure legends how the kanamycin cassette is removed (Page 8, lines 358-361). We would like to point out that a detailed description of en passant mutagenesis steps is provided in the methods section as well.

“…4th Step: In the second step of Red recombination I-SceI homing endonuclease expression by arabinose induction, mediated double-strand break and the seamless removal of the kanamycin cassette, resulting in BAC mutants carrying the corresponding spike mutations for B.1 and B.1.617.2 isolates…”

Line 311 : same comment for line 144 : until how many days ?

Response 9: We provide this information in the method section (see Response 7) and edited the text as follows (Page 7, lines 332-334):

“…Recombinant viruses WTBAC-V, B.1BAC-V and B.1.617.2BAC-V were reconstituted in Caco-2 cells and the supernatant were harvested and used for an additional infection round to generate passage one stocks…”

Line 313 : table S3 : confusing. Please indicate which mutations appeared unintentionally : during cell culture or after transfection for example (such as L84S for the WT-BAC-V).  

Response 10: For the generation of recombinant viruses, we used the SARS-CoV-2 BAC construct based on the USA-WA1/2020 strain which is named as WTBAC in this study (Page 3, lines 118-119). This parental strain has an L84S mutation when compared to Wuhan strain. We used Nanopore sequencing to check for additional mutations that might be introduced accidently by passaging of the viruses but did not observe any change.

In order to avoid misunderstanding we updated the former Table S3 (now Table S4) and edited the main text describing this mutation (Page 8, lines 343-345):

“…cDNA BAC clone of the SARS-CoV-2 USA-WA1/2020 isolate carrying a single amino acid substitution in ORF8 (L84S) and is represented as WTBAC-V…”

Line 336 : I feel really confused. The author replace the spike of the Wuhan strain by the one of B1. When they did it, they observed a higher viral fitness of the recombinant virus compare to the clinical B1 isolate itself? How is this possible ? The mutation in Table S3 that emerged P314L (I suppose because it’s not indicated) could affect this fitness ? Does the non-spike mutation negatively impact the viral fitness ? This results must be confirmed with the generation of a recombinant virus carrying only the non spike mutation !

Response 11: The onset of CPEs with the clinical isolates seem to be delayed by 4h in A549-AT cells (Figure 2B). We would like to avoid over interpretation of this observation as a sign of viral fitness, as we did not detect significant differences between clinical isolates and BAC viruses when analyzing the intracellular and extracellular RNA levels at 12 hpi (Figure 3) although TCID50/ml calculation in A549-AT infected cells at 24 hpi (Figure 2A) and human bronchial epithelial cells (Figure 7C) showed a tendency of increased infectivity. We elaborated on this issue in the Response 4 of major comments section and edited the text in the discussion (Page 16, lines 603-619)

“…On the basis of total N-gene (intracellular and supernatant) qRT-PCR, we detected sig-nificantly faster replication kinetic in cells infected with BAC-V at early time points (Table S2). Despite lower inoculum, RNA levels in CI-infected cells matched the BAC-V at 6-8 hpi time point. This correlates with the RNA levels detected in the supernatant (Figure 3). Since at 12 hpi time point the syncytia are still intact, it can be assumed that total N-gene RNA measured in the supernatant is mostly due to exocytosis and can therefore reflect virus production. Although not significant, we observed a tendency of increased progeny virus titers in A549-AT cells infected with the clinical isolates, especially B.1.617.2CI, at 24 hpi (Figure 2A, right). Primary HBEpCs infected with B.1.617.2CI showed significantly higher RNA levels in cell lysates and supernatant. Consistent with these observations, we detected higher infectious particles in the supernatant of the cells infected with, B.1.617.2CI at 5 dpi (Figure 7C). On the basis of these findings, we can suggest stronger CPE in HBEpCs and longer persisting syncytia in A549-AT infected with B.1.617.2CI may allow prolonged virus replication. Based on concordant observations from two different cell models, we carefully suggest a potential role of non-spike mutations in infection kinetics by either modulating syncytium formation and/or virus replication. In future studies, gene-by-gene molecular mapping of non-spike mutations by generating recombinant viruses would be critical to prove these observations...”

Line 362 : figure 3 : please indicate where are the significant differences in the figure.

Response 12: We performed Welch-t test and compared whether the differences in RNA levels detected from cells/supernatants from infected cells showed any significant differences (Table S2)

As we indicated in the manuscript we detected significant differences in total N-gene levels at 2 and 4 hpi time points which is likely due to the differences in the inoculum (0 hpi). We edited the text in the discussion (Page 16, Lines 600-610)

“…Although the RNA levels in the inoculum seem to slightly differ between CI and BAC-V (Table S2), back titration of the viruses showed that similar infectious particles were used (Table S5 B). This difference is possibly due to presence of unpacked viral RNA intermediates released into the supernatant upon lysis. On the basis of total N-gene (intracellular and supernatant) qRT-PCR, we detected significantly faster replication kinetic in cells infected with BAC-V at early time points (Table S2). Despite lower inoculum, RNA levels in CI-infected cells matched the BAC-V at 6-8 hpi time point. Since at 12 hpi time point the syncytia are still intact, it can be assumed that total N-gene RNA measured in the supernatant is mostly due to exocytosis and can there-fore reflect virus production. This correlates with the RNA levels detected in the supernatant (Figure 3). …”

Line 401 : figure 4 : my previous comment also come for here. For the 3x BNT results, NT50 for clinical and recombinant B1 or B1.617.2 were comparable. So we could speculate that the non spike mutation may not be relevant. However, this should be confirm by the respective recombinant virus carrying the non-spike mutation and confront to this result.  For the 2x MOD+BNT booster, the NT50 was higher for B1 BAC : what is the conclusion here? Does the non spike protein are involved and interfere with the neutralization ? Authors should at least discuss about this in the discussion section. From here, the authors didn't used the WT-BAC-V anymore. Why ? How can we be certain that the mutation recorded on this backbone (L84S) also present in the recombinant viruses was not involved in these testing?

Response 13: Since the Spike region is the main target of BNT and MOD vaccine-elicited antibody neutralization and we do not think it is necessary to generate non-spike mutation carrying recombinant viruses to show whether the P323L mutation in RdRp has any relevance. 2xMOD+BNT booster antibody show significant difference for B.1 but this might be due to vaccine combination used to elicit the antibody response. We purposefully avoided drawing concrete conclusions due to the limited sample size and its representation of general population. We sought to perform comparative virus neutralization using vaccine-antibody and mAbs in a well-controlled setting. We agree that NT analysis should be better discussed and included the following sentences in the discussion (Page 16, lines 627-634).

“…3xBNT response for both variants are comparable for clinical isolates and corresponding BAC variants. In contrast, B.1BAC-V, when compared to B.1CI, seemed to be neutralized significantly more efficiently by 2xMOD+BNT. Since the spike protein is the major target of vaccine-elicited antibodies tested in this study, the difference we observed in the NT assay is unlikely to be due to the non-spike mutation P323L in B.1CI. The superior breadth and potency of neutralizing antibodies due to heterologous combination might play a role, however, the sample size used for this experiment was limited, and therefore it is not possible to draw any concrete conclusions …”

We did not test WTBAC-V as we intended to compare recombinant viruses with their respective clinical isolate. WT BAC-V serves tough as a useful control when it comes to assessment of replication dynamics. To assess the replication dynamics, we included a virus that is closest to the original SARS-CoV-2 virus to obtain a baseline for our comparison. On the other hand, the inclusion of WT BAC-V into neutralization experiments with mAbs, vaccine sera or antivirals does not provide additional information, as we sought to compare variant-specific spike mutations between the BAC viruses and corresponding clinical isolates.

Line 464 : figure 7 : Not easy to read. All this graph can be place in a same pannel. This could help the reader to see/compare the different between mock-infected and infected cells.

Response 14: We thank the reviewer for pointing this out. We updated this figure in the manuscript. (Page 15, Line 552)

Line 575 : in lines 423-424 authors claimed that "treatment with Imdevimab displayed diminished protection against both recombinant variants, B.1BAC-V and B.1.617.2BAC-V". This mean that non spike mutation play a role. Since the two mAB used here targeted the spike RDB, authors should clairified the implication of the non spike mutation in the neutralization

Response 15: We appreciate the reviewer’s suggestion. In the paragraph in lines 636-646, we discuss the obtained results for mAB treatment and possible reasons for the observed differences between clinical isolate and recombinant virus while considering the previous gained knowledge tied to our infection kinetic results. Non-spike mutations seem to influence CPE formation and persistence and therefore control the release of virus during the infection process. Differences are hence not of mechanistic origin regarding neutralization by mAbs but rather reflect different replication dynamics by the viruses.

Reviewer 2 Report

SARS-CoV-2 is a constantly evolving virus with mutations in the receptor-binding site, most notably D614G, known to be major determinants of increased virulence, evidenced by the emergence of the B.1 variant.  A subsequent variant, B.1.617.2 (Delta), amassed additional spike mutations, resulting in markedly enhanced transmissibility and the possible ability to escape vaccine protection.  In addition to spike mutations, substitutions have been identified in non-spike regions of these variants, including the nucleocapsid and non-structural proteins, that have been shown to increase viral fitness.  In this study, the goal is to understand the relative contributions of spike protein mutations and non-spike mutations to the viral life cycle, especially replication dynamics and the development of cytopathic effects and sensitivity to neutralizing antibodies and antiviral therapies.  In order to address these issues, recombinant SARS-CoV-2 mutants were constructed carrying the B.1 or Delta spike domains and their critical mutations in the backbone of the SARS-CoV-2 USA-WA1/2020 clinical isolate.  These recombinant viruses were compared to the B.1 and Delta variants and the clinical isolate with respect to the parameters above.

The data reveal no significant differences between the recombinant viruses and the isolates with respect to their susceptibilities to neutralization by vaccine and convalescent sera or anti-spike monoclonal antibodies.  Similarly, the antiviral efficacies of several antivirals, including Remdesivir and Molnupiravir, were comparable against the recombinant viruses and the clinical isolates.  Thus, non-spike mutations for either B.1 or Delta did not appear to affect the sensitivity of the virus to antibody neutralization or inhibition by antivirals. 

With no discernible effect on antibody or antiviral sensitivity, the authors undertook a comparison of the replication properties of the recombinant viruses and clinical isolates.  To do so, differentiated human bronchial epithelial cells were infected and analyzed for cytopathic effects, barrier integrity and RNA expression.  They observed enhanced CPE formation such as persisting syncytia with the recombinants relative to the parent viruses.  In keeping with this observation, an impairment of the epithelial barrier, especially in the Delta recombinant, was detected by measurement of transepithelial electrical resistance (TEER).  Based on analyses of RNA production, the authors concluded that the non-spike mutations may enable a prolonged duration of virus replication, in turn, possibly affecting virus production.

This is considered a very technologically-sophisticated, competently executed and appropriately interpreted study.  The authors are to be commended for their attention to detail, careful interpretation of the data and for not overstating the significance of their findings.  That being said, the primary contribution made by the study to our understanding of SARS-CoV-2 is really only suggestive of a potential role for non-spike mutations in infection kinetics.  Although it is understandably likely beyond the scope of this study, it would be highly informative to determine the effect of non-spike mutations on the virulence, pathogenicity and transmissibility of the virus.

Minor point

The authors should comment on the difference in neutralization efficacy against recombinant and clinical isolates obtained with the 3X BNT and 2X MOD+ BNT boosters.

Author Response

Response to reviewer 2

Comments and Suggestions for Authors

SARS-CoV-2 is a constantly evolving virus with mutations in the receptor-binding site, most notably D614G, known to be major determinants of increased virulence, evidenced by the emergence of the B.1 variant.  A subsequent variant, B.1.617.2 (Delta), amassed additional spike mutations, resulting in markedly enhanced transmissibility and the possible ability to escape vaccine protection.  In addition to spike mutations, substitutions have been identified in non-spike regions of these variants, including the nucleocapsid and non-structural proteins, that have been shown to increase viral fitness.  In this study, the goal is to understand the relative contributions of spike protein mutations and non-spike mutations to the viral life cycle, especially replication dynamics and the development of cytopathic effects and sensitivity to neutralizing antibodies and antiviral therapies.  In order to address these issues, recombinant SARS-CoV-2 mutants were constructed carrying the B.1 or Delta spike domains and their critical mutations in the backbone of the SARS-CoV-2 USA-WA1/2020 clinical isolate.  These recombinant viruses were compared to the B.1 and Delta variants and the clinical isolate with respect to the parameters above.

The data reveal no significant differences between the recombinant viruses and the isolates with respect to their susceptibilities to neutralization by vaccine and convalescent sera or anti-spike monoclonal antibodies.  Similarly, the antiviral efficacies of several antivirals, including Remdesivir and Molnupiravir, were comparable against the recombinant viruses and the clinical isolates.  Thus, non-spike mutations for either B.1 or Delta did not appear to affect the sensitivity of the virus to antibody neutralization or inhibition by antivirals. 

With no discernible effect on antibody or antiviral sensitivity, the authors undertook a comparison of the replication properties of the recombinant viruses and clinical isolates.  To do so, differentiated human bronchial epithelial cells were infected and analyzed for cytopathic effects, barrier integrity and RNA expression.  They observed enhanced CPE formation such as persisting syncytia with the recombinants relative to the parent viruses.  In keeping with this observation, an impairment of the epithelial barrier, especially in the Delta recombinant, was detected by measurement of transepithelial electrical resistance (TEER).  Based on analyses of RNA production, the authors concluded that the non-spike mutations may enable a prolonged duration of virus replication, in turn, possibly affecting virus production.

This is considered a very technologically-sophisticated, competently executed and appropriately interpreted study.  The authors are to be commended for their attention to detail, careful interpretation of the data and for not overstating the significance of their findings.  That being said, the primary contribution made by the study to our understanding of SARS-CoV-2 is really only suggestive of a potential role for non-spike mutations in infection kinetics.  Although it is understandably likely beyond the scope of this study, it would be highly informative to determine the effect of non-spike mutations on the virulence, pathogenicity and transmissibility of the virus.

Response 1: We would like to thank the reviewer for the comments.

We concur that it is critical to investigate mutational landscape in other regions and its role in virus life cycle. In this current study we focused on the spike region due to its importance for therapeutic interventions, vaccine design and efficacy. Based on our findings we now have started looking into the role of Nucleocapsid and RdRp regions using en passant mutagenesis. We seek to understand the role of individual mutations or the mutations in concert for virus replication gene by gene, especially for more recent VoCs. However, due to the pace of research and publication rate in this field, we prefer sharing our results with the community in a timely manner.

Minor point

The authors should comment on the difference in neutralization efficacy against recombinant and clinical isolates obtained with the 3X BNT and 2X MOD+ BNT boosters.

Response 2:  We edited the discussion and included the following section (Page 16, lines 627-634).

‘…3xBNT response for both variants are comparable for clinical isolates and corresponding BAC variants. In contrast, B.1BAC-V, when compared to B.1CI, seemed to be neutralized significantly more efficiently by 2xMOD+BNT. Since the spike protein is the major target of vaccine-elicited antibodies tested in this study, the difference we observed in the NT assay is unlikely to be due to the non-spike mutation P323L in B.1CI. The superior breadth and potency of neutralizing antibodies due to heterologous combination might play a role, however, the sample size used for this experiment was limited, and therefore it is not possible to draw any concrete conclusions …”
